# Fertility Life Table, Thermal Requirements, and Ecological Zoning of *Anthonomus grandis grandis* Boheman (Coleoptera: *Curculionidae*) in Brazil

**DOI:** 10.3390/insects14070582

**Published:** 2023-06-26

**Authors:** Fernanda Polastre Pereira, Alexandre José Ferreira Diniz, José Roberto Postali Parra

**Affiliations:** 1Department of Entomology and Acarology, “Luiz de Queiroz” College of Agriculture, University of São Paulo (USP), 11 Pádua Dias Ave, Piracicaba, Sao Paulo 13418-900, Brazil; jrpparra@usp.br; 2Department of Plant Protection, Rural Engineering and Soils (DEFERS), São Paulo State University (UNESP), 56 Brasil Sul Ave, Ilha Solteira, Sao Paulo 15385-000, Brazil; jose.diniz@unesp.br

**Keywords:** biology, boll weevil, temperature, cotton, pest zoning

## Abstract

**Simple Summary:**

The boll weevil is a key pest of cotton in Central and South America. Its thermal requirements, development time, oviposition, survival, adult longevity, and sex ratio were determined under laboratory conditions, and the data were used to construct a fertility life table. Based on the results and a GIS (Geographic Information System), the R_0_ was estimated for different Brazilian regions and represented on a map. This information can be useful for developing strategies to manage the pest in cotton crops.

**Abstract:**

The boll weevil, *Anthonomus grandis grandis* Boh., is the most important cotton pest in Central and South America. The biological characteristics and thermal requirements of boll weevils reared on an artificial diet were assessed at seven constant temperatures (18, 20, 22, 25, 28, 30, and 32 ± 1 °C) under laboratory conditions. These data were used to determine the ecological zoning for the pest in Brazil. The development time; oviposition period; the number of eggs produced; survival of eggs, larvae, and pupae; adult longevity; and sex ratio were recorded, and additional life table parameters were calculated. The total development duration ranged from 16.1 (32 °C) to 46.2 (18 °C) days. Temperature significantly affected the number of eggs laid per female (fecundity), with the highest number of eggs observed at 25 °C (251 ± 15.8). The parameters from the fertility life table indicated the greatest population growth at 25 °C and 28 °C. The net reproductive rate (R_0_) at these temperatures was 22.25 times higher than at 18 °C. Based on R_0_ and temperature, an ecological zoning of the pest was developed for Brazil. Brazilian regions with mean temperatures above 20 °C and below 30 °C are most favorable for the population growth of the boll weevil. The most suitable crop areas were found to be the north, midwest, and part of the northeast region, although the weevil can occur throughout Brazil if the host plants are available.

## 1. Introduction

The cotton boll weevil is the most important pest of cotton crops in Brazil due to the damage it causes and the difficulty of its control; when no measures are taken to control the pest, the boll weevil can destroy the production of a cotton field. This pest has a high capacity to damage cotton fruiting structures due to its short life cycle and high capacity for population growth and dissemination in the field. This insect was described by C. H. Boheman in 1843 and was reported as originally from Mexico, from where it spread to the United States, invading the state of Texas in 1892. Within a few years, it spread through the southwestern United States, covering a large part of the cotton belt, with serious economic and social consequences. The weevil has now spread to countries in Central and South America [1,2,3].

In Brazil, the boll weevil was first reported in February 1983 in Campinas, São Paulo [4]. Burke et al. [3] determined that the boll weevil was introduced into Brazil from the United States. Additionally, in 1983, the weevil was recorded in another 46 municipalities in the state of São Paulo, as well as states in the northeastern and southern regions [5]. The boll weevil has spread and is established in all cotton-growing regions of the country, including the Cerrado and Caatinga biomes. The cotton ecosystem in Brazil is highly favorable to boll weevils, with suitable weather during the crop season and available food to support during the fallow period. As a result, local cotton crops have suffered significant damage, requiring an increased number of insecticide applications to control the pest [6,7,8].

In Brazil, in the 2021–22 crop season, approximately 1.6 million hectares were planted, mainly in Mato Grosso and Bahia [9]. Costs to growers for boll weevil management are estimated at USD 140 to 350 per hectare, besides indirect costs due to disruptions of IPM programs and environmental issues [8,10]. In Brazil, *A. grandis grandis* is the main cotton pest, as it shows high survival and reproductive rates, adaptation, dispersal, and migration in different environments due to the favorable climatic conditions in the region [11,12]. During the fallow period, when cotton plant residues are removed in compliance with legislation, boll weevils are found around the edges of the cotton fields, mainly in facultative reproductive dormancy. At this time, females do not produce eggs, and males have testicular atrophy [13]. Adult boll weevils can survive the fallow season by feeding on nectar and pollen from different native plants [14].

Mathematical models are commonly used to predict the occurrence of agricultural pests [15,16,17]. Among the components of a model, temperature is one of the most important climatic elements for the development and survival of many insect species [18]. Knowledge of the thermal requirements of the different stages makes it possible to better understand the population dynamics, predict the occurrence of pests in crops, increase sampling efficiency, and optimize control strategies. The thermal requirements of the insect are assessed using the thermal constant (K), expressed in degree days, which has been used for many years in studies predicting plant growth. The hypothesis is that the duration of development, influenced by temperature, is constant, with the sum of temperatures calculated above the temperature development threshold (Tb) [19,20,21]. Since insects are poikilothermic, i.e., adapt to the ambient temperature, this thermal constant also applies to their development [18]. Combining life table data with the study of thermal requirements makes it possible to show the patterns of survival and reproduction of a given population at different temperatures [22,23,24,25]. Additionally, GIS (Geographic Information System) modeling allows the representation of the relationship between temperature and spatial distribution [26]. GIS users can computationally analyze geospatial data, including collection, analysis, manipulation, and representation of the data set [27]. This information is essential to monitor and control a pest in the context of integrated pest management [28].

Maintaining populations on natural hosts is essential for certain insect groups; however, it requires a significant amount of labor, and maintenance is usually challenging as it depends on the region, planting time, and need for facilities such as greenhouses or controlled environments (temperature, relative humidity, and photoperiod). Rearing populations on artificial diets offers several advantages, including less labor and ease of population control. This allows the maintenance of insect populations in the laboratory and enables the continuity of research in the field. Artificial diets also enable the production of insects on a small scale, such as in taxonomic studies, bioassays, and evaluation of nutritional quality, as well as on a large scale, such as in mass rearing of parasitoids and predators, use of genetic technologies, and production of pheromones [29,30].

In 1958, Vanderzant and Davich developed the first artificial diet for the cotton boll weevil to optimize laboratory rearing [31]. Subsequent studies improved the diet’s nutritional quality, including that of Sterling and Adkisson [32], which yielded equal or superior results for female fecundity, using a cottonseed meal diet to feed both larvae and adults. In 1979, Lindig tested Pharmamedia^®^ (ADM, Chicago, IL, USA), a commercially available cottonseed protein, as a substitute for cottonseed meal [33]. The most recent artificial diet was proposed in 2000 by Monnerat et al. [34], based on brewer’s yeast, wheat germ, soy protein, Pharmamedia^®^ cottonseed protein concentrate, Wesson salt, sugar, ascorbic acid, sorbic acid, methyl parahydroxybenzoate, vitamin solution, agar, and distilled water [34,35].

In this study, we investigated the development of *A. grandis grandis* reared on an artificial diet at seven different temperatures, determined its thermal development limits, and generated a fertility life table. These data were applied to Geographic Information System (GIS) tools to identify the most suitable areas for *A. grandis grandis* based on temperature. This information can aid in identifying potential areas for the expansion and occurrence of this pest and in developing strategies for managing it in cotton crops.

## 2. Materials and Methods

### 2.1. Insect Rearing

Adults of *A. grandis grandis* were obtained from cotton plants with larvae-infested squares (flower buds) in an experimental field of the Entomology Department (USP-ESALQ), Piracicaba, São Paulo, Brazil (22°43′30′′ S/47°38′56′′ W). The experiment was conducted at the Laboratory of Insect Biology, USP—ESALQ. The boll weevil stock colony was reared on an artificial diet adapted from [34], based on brewer’s yeast, wheat germ, soy protein, Pharmamedia^®^ (ADM, USA), concentrated cottonseed protein, Wesson salt, sugar, ascorbic acid, sorbic acid, methyl parahydroxybenzoate, vitamin solution, agar, and distilled water. The insects were maintained under controlled conditions (25 ± 2 °C, 70 ± 10% RH, and a photoperiod of 14:10 [L:D]).

Adults were placed in plastic cages (20 × 15 × 10 cm) with the center of the lids removed and replaced with screens (Mesh 60). At the bottom (base), another screen (Mesh 120) was placed to allow frass (feces) and eggs to pass into an unscreened plastic container of the same dimensions below the cage. Boll weevil eggs were collected with feces and deposited into a solution of 20% copper sulfate. In this solution, the eggs float while the feces precipitate. The eggs were collected, disinfected for 1 min with 0.3% benzalkonium chloride, and inoculated into Petri dishes (60 mm × 15 mm) containing 30 mL of the same diet provided to the adults. The immature stages developed in these dishes until the emergence of adults, which were collected and transferred to screened cages. The plastic pots were previously sterilized with 70% ethanol and placed under a germicidal lamp for 15 min. With the aid of a thick brush, Vaseline was applied around the edge of the container to prevent the insects from escaping during handling. The adults were offered pieces of the artificial diet and maintained in these cages for 35 to 40 days. Three times a week, eggs, feces, and dead insects were removed, and the artificial diet was replaced.

### 2.2. Biology of A. grandis grandis at Different Temperatures on Artificial Diet

Seven Biochemical Oxygen Demand (BOD) climate chambers (Electrolab) were programmed with different temperatures (18, 20, 22, 25, 28, 30, and 32 ± 1 °C), 70 ± 10% RH, and a photoperiod of 14:10 [L:D], with each temperature corresponding to a treatment. Temperature and relative humidity (RH) were monitored using a Gemini Tinytag Ultra 2 Datalogger. This temperature regime corresponds to most of the monthly mean temperatures in different Brazilian regions. Boll weevil eggs up to 24 h old were placed in Petri dishes (60 mm × 15 mm) containing moistened qualitative filter paper, 80 g (in 5 replicates of 50 eggs each). Two hundred fifty eggs were maintained at each temperature. Development (egg-larva) and egg survival were recorded daily to assess duration and viability. For the analysis of larva-pupa and pupa-adult survival and development duration (larva-adult), larvae that hatched from the Petri dishes were inoculated with a fine brush into wells in Greiner Bio-One Polystyrene 24-well Cell Culture Multiwell Plates, each well containing one larva and 2 mL of artificial diet. For each of the seven temperatures (treatments), five replicates were run, each replicate consisting of 24 insects, totaling 120 insects per temperature. The larvae were observed daily to determine how many reached the pupal stage. The survival of pupae to adults and the sex ratio was determined by counting the males and females that emerged from pupae held in Greiner Bio-One Polystyrene 24-well Cell Culture Multiwell.

Twenty-four hours after the emergence, all the weevils were separated by sex according to the description in Sappington and Spurgeon [36]. Then, 25 males and females were paired and reared in cages composed of an inverted 500 mL plastic cup fitted over a plastic lid. On the upper surface, 10 holes were drilled to prevent an increase in relative humidity in the cups. Two pieces (1.0 × 1.0 × 0.3 cm) of artificial diet were provided as food to the adults in each cup, and the artificial diet was replaced daily.

### 2.3. Statistical Analysis

#### 2.3.1. Biology

The following parameters were calculated: duration of development (egg-adult); survival of eggs, larvae, and pupae; pre-oviposition (time period before females begin to lay eggs); the number of eggs laid (fecundity); adult longevity; and sex ratio. All data were tested for homogeneity [37], normality, and independence of the residuals [38]. Data were analyzed using generalized linear models (GLM) with Poisson distribution for the mean duration of adult longevity, pre-oviposition, and fecundity. Data on survival and sex ratio at the different temperatures were analyzed using GLM with Binomial distribution, followed by Tukey’s multiple comparison tests with a 95% confidence index (GLHT package) [39,40]. The relationship between the development time of each stage and the temperatures was described using linear regression with the lm package in R [40], with development time as the dependent variable and temperature as the independent variable. The performance of the linear model to fit the data was tested using the coefficient of determination (r2).

#### 2.3.2. Thermal Requirements and Fertility Life Table

After the durations of the development stages at different temperatures were determined, the lower threshold temperature (Tb) and thermal constant (K) were calculated using the following linear equation [19]: 1/D = a + bT, where 1/D is the development rate (d–1), and T is the temperature (°C). The lower threshold temperature T was calculated as the ratio between the angular and linear coefficients of the line (−a/b), and the thermal constant (K) was obtained using the quotient (1/b) [20,21]. 

The life history data on *A. grandis grandis* were constructed using the age-specific fecundity, number of eggs produced daily by one female, age-specific fecundity of the total population, age-specific development, and age-specific survival rate. The following parameters were determined: R_0_, net reproductive rate; T, duration of each generation; rm, intrinsic rate of increase; λ, finite rate of increase; and Dt, generation doubling time. TWOSEX-MSChart software [41] was used to calculate the parameters of the life table. Statistical comparison of values was performed using the Bootstrap test, available in the TWOSEX-MSChart software [41,42,43].

#### 2.3.3. Occurrence of *A. grandis* in Brazil Based on Fertility Life Table

In examining the relationship between R_0_ and the temperature treatments, six different nonlinear models were tested: Lactin-1, Lactin-2, Logan-6, Logan-10, Gaussian, and a second-order polynomial model. The parameters of the nonlinear models were estimated using the minpack.lm package from R [40]. The models were evaluated based on the coefficient of determination (R^2^), adjusted coefficient of determination (R^2^_adj_), residual sum of squares (RSS), and corrected Akaike Information Criterion (AICc) [21].

The mean annual temperatures at 296 georeferenced locations in Brazil [44] and the equation that best fits the observed relationship between R_0_ and temperature were used to estimate the net reproductive rate (R_0_) for each location. The QGIS software was used to organize and represent the values of R_0_ in shapefiles. Then, IDW (Inverse Distance Weighting) interpolation was applied to the R_0_ values for each location, providing a visual description of the weevil reproductive rates in Brazil [45]. The cotton-producing regions of the country are highlighted on the map. These were chosen based on the Market Report for cotton production systems in the Brazilian Agricultural Production survey [46].

## 3. Results

### 3.1. Biology of A. grandis grandis at Different Temperatures

The boll weevils completed development at all temperatures used in this experiment (18 to 32 °C), and the development time of each stage shortened as the temperature increased. The longest and shortest embryonic periods were 6.2 and 2.2 days at 18 °C and 32 °C, respectively (Figure 1). The larval stage was 20.1 days longer at 18 °C than at 32 °C. The longest pupal stage was observed at 18 °C (10.50 days) and the shortest (4.19 days) at 32 °C. The total duration of development was longest (46.2 days) at 18 °C and shortest (16.1 days) at 32 °C, and differed significantly among all temperatures (Figure 1). Survival differences were observed as a function of temperatures for eggs (F_6, 28_ = 20.45, *p* < 0.001) and larvae (F_6, 28_ = 7.10, *p* < 0.001), but not for pupae (F_6, 28_ = 2.17, *p* = 0.07). Survival of eggs was lower at 18 °C (45%), 20 °C (54.2), and 32 °C (52.5), indicating that the egg stage was more vulnerable at high and low temperatures. Egg survival was highest at 25 °C (75.8%). Larvae showed the same pattern, with the lowest survival at 18 °C (68.4%) and 32 °C (66.4%), differing from the other temperatures. Survival of pupae did not differ significantly among the temperatures tested. The values recorded for the survival of eggs and larvae indicate that the most suitable temperature range for these stages is between 25 and 28 °C (Table 1).

The pre-oviposition period decreased significantly from 10.2 ± 0.13 to 3.4 ± 0.18 days as the temperature increased from 18 to 32 °C (F_6, 134_ = 254.9, *p* < 0.001) (Table 2). The same pattern was observed for adult longevity; females (F_6, 136_ = 137.1, *p* < 0.001) and males (F_6, 164_ = 219.4, *p* < 0.001) lived longer at 18 °C than at the other temperatures tested (Table 2). The results for increased longevity at lower temperatures were expected, as metabolic activity is lower at these temperatures. Temperature did not affect the sex ratio (ca. 0.43 to 0.51) (F_6, 28_ = 0.21, *p* = 0.97) (Table 2). 

However, fecundity varied among temperatures (F_6, 132_ = 28.14, *p* < 0.001). Fecundity was lowest at 32 °C and 18 °C, and highest at 25 °C (Table 2). The appropriate temperature regime for boll weevil egg-to-adult development, survival, and fecundity was 25 °C to 28 °C (Figure 1, Table 1 and Table 2).

### 3.2. Thermal Requirements and Fertility Life Table of A. grandis grandis on Artificial Diet at Seven Constant Temperatures

The calculated Tb and K were 10.18 °C and 357.84 DD. The coefficient of determination was higher than 95% (ca. 0.96) for the linear model fitted and used to estimate K and Tb. Based on the parameters of the fertility life table constructed with different temperatures, the highest reproductive performance was between 25 and 28 °C (Table 3). The R_0_ differed among all temperatures, except within this favorable temperature span. The mean generation time (T) decreased significantly as the temperature increased (Table 3). The highest intrinsic rate of increase (rm) value was 0.105 femalexfemalexday at 25 °C and 28 °C, which differed significantly from the rates estimated at other temperatures. The finite rate of increase (ʎ) showed a similar trend, with a value of 1.111 at 25 °C and 28 °C (Table 3).

### 3.3. Occurrence of A. grandis grandis Based on Fertility Life Table in Cotton Fields in Brazil

Of the mathematical models used to examine the relationship between temperature and R_0_, the Gaussian model (Figure 2) showed the best fit, as evidenced by higher values of the coefficient of determination (R^2^) and adjusted coefficient of determination (R^2^_adj_), and lower values of the residual sum of squares (RSS), and corrected Akaike Information Criterion (AICc) [21].

Based on the net reproductive rate (R_0_), an interpolated map of pest distribution was developed, taking temperature into account (Figure 3). The temperature map showed that the highest values (R_0_ ≥ 70) are concentrated in the north, northeast, and midwest regions (Figure 3). The map (Figure 3) shows that in the cotton-producing areas of the midwest and northeast regions (where the major cotton-producing states of Mato Grosso and Bahia are located), population growth is higher. Additionally, the mean temperatures during the fallow season in these regions, together with the biological characteristics of the insect, contribute to the development of the pest in these areas.

## 4. Discussion

Comparative biological aspects and demographic parameters were assessed through the fertility life table of the cotton boll weevil at seven constant temperatures. The results indicated a significant interaction between temperature and boll weevil in developmental duration and adult performance. Temperature and development were negatively related to the temperature regime studied (18–32 °C) for embryonic and post-embryonic development and adult traits. The total duration of development differed significantly among all temperatures, ranging from 16.10 (32 °C) to 46.20 (18 °C) days. This pattern agrees with the results of previous studies evaluating the development of this species in different temperature conditions [47,48,49]. Both embryonic and larval survival were affected by extreme temperatures, being lowest at 18 °C (45 and 68.4%, respectively) and 32 °C (52.5 and 66.4%, respectively). Similarly, Cole et al. [48], evaluating the survival of boll weevils over the range of 15.6 to 32.2 °C, indicated optimum temperatures for embryonic and larval development of around 23.9 °C and 29.5 °C, respectively. As temperatures moved toward extremes, the percentage of survival decreased. Greenberg et al. [49] observed the highest survival of boll weevil immature stages at 25 °C (63.6%), in agreement with our data. 

The highest fecundity was found at 25 °C and 28 °C, close to the range found by Greenberg et al. [49], who reported female fecundity rates of 185 eggs per female at 25 °C. Likewise, Cole et al. [50] reported that the highest number of eggs was produced at 29.5 °C (253.5 eggs). Lack of oviposition was observed at 15.6 °C, and temperatures above 29.5 °C by Greenberg et al. [49] had a negative impact on oviposition. The longevity of *A. grandis grandis* estimated here was much higher than that observed by Greenberg et al. [49], who found a longevity of 36 days for females at 25 ± 1 °C. The higher longevity of adults observed in this study may indicate that the use of diets based on different components may be responsible for the differences in longevity in these studies, since nutrition affects the development of the insect [51]. The present results for boll weevil biology indicate that the artificial diet is suitable and can replace the natural diet for rearing *A. grandis grandis* in the laboratory, aiming at the production of natural enemies.

This is the first study of the thermal requirements of the boll weevil in Brazil. The lower threshold temperature for *A. grandis grandis* estimated here (10.18 °C) is close to the estimate of Fye et al. [52], differing by approximately 2 °C. The fertility life table constructed at different temperatures is useful for understanding the population dynamics of target species due to the integration of biological variables of insects [29]. Population growth parameters obtained using the fertility life table indicated optimum growth at 25 °C or 28 °C. *A. grandis grandis* had the highest intrinsic rate of increase (rm) and finite rate of increase (ʎ) in this thermal range at 25 °C and 28 °C, which was significantly different from the values observed at other temperatures. At these temperatures, the reproductive rate R_0_ was 22.25 times higher than at 18 °C. This optimal temperature range for population growth was also observed by Greenberg et al. [49], i.e., an R_0_ value of 66.8 at 25 °C, close to that found in this study (74.77). 

The thematic map based on R_0_ and temperature indicates that regions with a tropical climate where mean temperatures are above 20 °C are more favorable for populations of *A. grandis grandis*. In these regions, which correspond to the Brazilian Midwest and Northeast (and a small portion of the southeast) regions, the estimated R_0_ was higher than 70 at 25 °C. This distribution agrees with findings by Azambuja and Degrande [12], who reported a strong presence of *A. grandis grandis* in the states of Bahia, Ceará, Paraiba, Pernambuco, and Rio Grande do Norte (northeast); Federal District, Goiás, Mato Grosso, and Mato Grosso do Sul (midwest); and São Paulo (southeast). The zoning presented here was based on the fertility life table parameters of *A. grandis grandis* at different temperatures. This method allows more biological traits to be taken into consideration, because R_0_ was used to develop the equation that estimates the distribution of *A. grandis grandis* in different Brazilian regions. Unlike methods that rely solely on thermal requirements, which estimate the number of generations per year in a given area, our approach includes additional biological data such as fertility and survival rates that directly impact population growth. However, the method disregards other physiological and behavioral factors in field conditions, such as the use of alternative food resources during the off-season [14,53]. In addition to the suitability of climate conditions, the availability of resources must also be taken into consideration regarding the establishment and maintenance of insect pests. According to a survey by Degrande et al. [11], the cotton boll weevil occurs in all crop areas shown in Figure 3. Although the lower threshold temperature was found at 10.18 °C, the insect, as reported in the literature, may undergo diapause or reproductive dormancy [13,54,55,56], which allows it to survive colder temperatures. We have observed that the average temperatures across all regions of Brazil indicate a value of $R_{0}$ above 5 for all regions, indicating that this species is able to thrive from the extreme south to the extreme north when the host plants are available.

In general, the method proposed here is useful for understanding the population dynamics of a species of interest, since it integrates the biological and ecological variables of insects with their geographic distribution. In summary, the information generated on the biology of the pest demonstrated the effect of temperature on its development and reproductive aspects. However, further studies are warranted to investigate additional abiotic factors such as relative humidity and precipitation, as well as a wider range of temperatures and the effect of fluctuating temperatures that might also influence the development and survival of *A. grandis grandis*. 

## 5. Conclusions

The present study found that temperature had a significant impact on the development, survival, fecundity, and longevity of *A. grandis grandis.* The parameters from the fertility life table indicated the highest population growth at 25 °C and 28 °C, with a thermal constant of 357.84 degree days, and a lower thermal development threshold of 10.18 °C. The pest zoning for Brazil, based on thermal requirements and a fertility life table, showed that the most favorable regions for the development and population growth of *A. grandis grandis* are the north, northeast, and midwest. These findings provide valuable guidance for effectively managing *A. grandis grandis* populations in the field.

## Figures and Tables

**Figure 1 insects-14-00582-f001:**
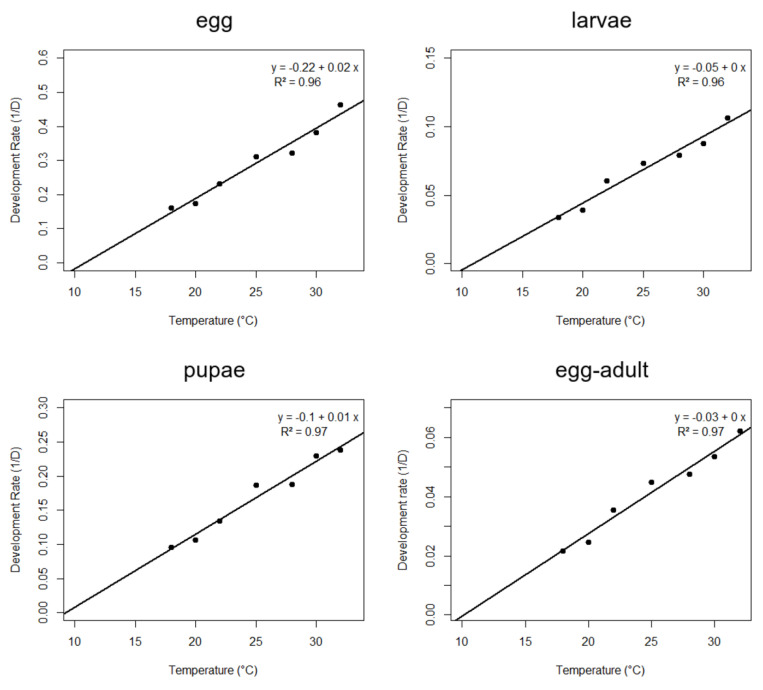
The temperature–dependent development rate of *A. grandis grandis* at different temperatures. RH 70 ± 10% and photophase of 14 [L:D].

**Figure 2 insects-14-00582-f002:**
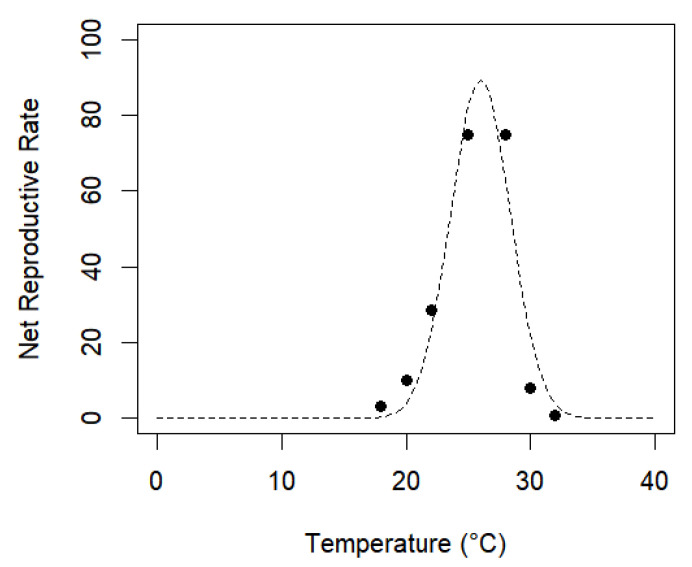
The Gaussian curve of the net reproductive rate (R_0_) as a function of tested temperature regimes for *A. grandis grandis*.

**Figure 3 insects-14-00582-f003:**
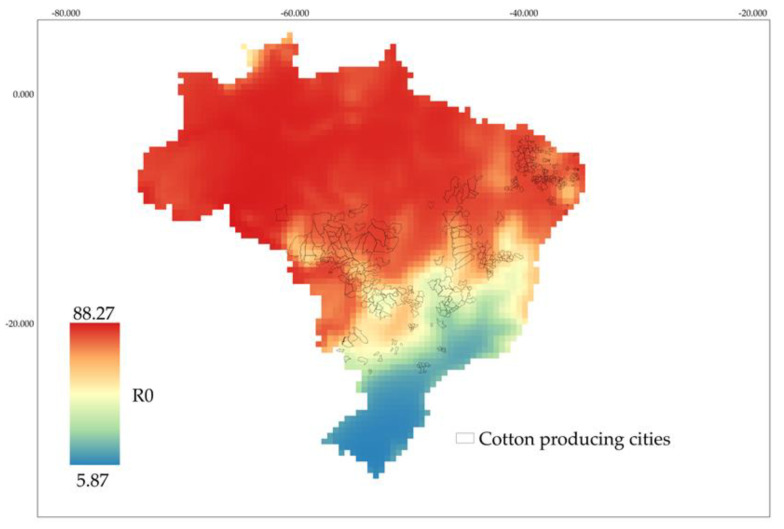
Interpolated map showing the estimated net reproductive rate of the cotton boll weevil based on temperature conditions. Black contour lines indicate cotton–growing regions of Brazil.

**Table 1 insects-14-00582-t001:** Survival (%) of different life stages of *A. grandis grandis* was recorded at seven constant temperatures (RH = 70 ± 10%; 14:10 [L:D]).

Temperature (°C)	Survival (%)			
	Egg	Larva	Pupa	Total (Egg-Adult)
18	45.0 ± 0.8 d	68.4 ± 7.5 c	75.7 ± 8.1 a	23.3 ± 2.1 c
20	54.2 ± 1.9 cd	72.8 ± 7.2 bc	75.1 ± 5.6 a	29.2 ± 4.5 c
22	65.0 ± 3.4 b	89.3 ± 2.0 b	85.5 ± 3.4 a	49.6 ± 4.5 ab
25	75.8 ± 3.1 a	95.4 ± 2.2 a	84.1 ± 5.2 a	60.8 ± 6.8 a
28	68.3 ± 2.1 ab	90.4 ± 4.1 ab	83.6 ± 1.9 a	51.6 ± 3.1 ab
30	59.2 ± 0.8 bc	77.3 ± 3.6 bc	81.3 ± 5.7 a	37.2 ± 3.9 bc
32	52.5 ± 2.2 cd	66.4 ± 3.4 c	63.9 ± 3.7 a	22.1 ± 2.5 c

Means followed by the same letter within a column do not differ from one another by Tukey’s test (α = 0.05).

**Table 2 insects-14-00582-t002:** Mean duration (±SE) of the pre-oviposition period, fecundity (eggs per female), sex ratio, and longevity of *A. grandis grandis* at seven constant temperatures (RH = 70 ± 10%; 14:10 [L:D]).

Temperature (°C)	Pre-Oviposition (Days)	Fecundity	Sex Ratio	Longevity (Days)	
				Females	Males
18	10.2 ± 0.13 a	33.7 ± 3.1 ab	0.51 ± 0.04 a	115.0 ± 2.8 a	111.0 ± 4.9 a
20	6.50 ± 0.15 b	99.3 ± 10.6 cd	0.43 ± 0.09 a	72.9 ± 2.2 b	66.5 ± 1.9 b
22	6.10 ± 0.13 b	156.0 ± 12.1 d	0.43 ± 0.07 a	75.2 ± 1.9 b	64 ± 3.1 b
25	5.56 ± 0.08 c	251.0 ± 15.8 e	0.50 ± 0.04 a	56.1 ± 1.4 c	55.1 ± 1.3 c
28	4.93 ± 0.04 d	234.0 ± 17.9 e	0.45 ± 0.05 a	47.7 ± 1.4 d	41.0 ± 1.9 d
30	4.52 ± 0.11 d	70.2 ± 9.6 bc	0.51 ± 0.10 a	34.4 ± 1.2 e	31.9 ± 1.4 e
32	3.38 ± 0.18 e	9.0 ± 1.5 a	0.49 ± 0.06 a	14.2 ± 1.3 f	11.8 ± 0.8 f

Means followed by the same letter within a column do not differ from one another by Tukey’s test (α = 0.05).

**Table 3 insects-14-00582-t003:** Life table and fertility of *A. grandis grandis* at seven temperatures (mean ± SE) (RH 70 ± 10% and photophase of 14:10 [L:D]). R_0_—net reproductive rate; rm—intrinsic rate of increase; λ—finite rate of increase; T—the mean period over which progeny are produced. Means followed by the same letter within a column do not differ from one another by Bootstrap test (α = 0.05).

T (°C)	R0	T	rm	ʎ
18	3.367 ± 0.972 a	72.948 ± 1.613 a	0.016 ± 0.004 a	1.016 ± 0.004 a
20	9.932 ± 2.896 b	64.874 ± 1.035 b	0.034 ± 0.005 b	1.035 ± 0.005 b
22	28.623 ± 5.921 c	50.782 ± 0.850 c	0.065 ± 0.004 c	1.067 ± 0.004 c
25	74.766 ± 11.411 d	40.754 ± 0.521 d	0.105 ± 0.004 d	1.111 ± 0.004 d
28	74.722 ± 11.446 d	40.755 ± 0.528 d	0.105 ± 0.004 d	1.111 ± 0.004 d
30	11.681 ± 2.841 e	30.179 ± 0.568 e	0.084 ± 0.008 b	1.083 ± 0.008 c
32	0.675 ± 0.239 f	22.419 ± 0.989 f	−2.084 ± 0.018 e	0.979 ± 0.001 e

## Data Availability

The datasets generated and/or analyzed in the current study are available from the corresponding author upon reasonable request.

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
