# Peer review of "Fertility Life Table, Thermal Requirements, and Ecological Zoning of *Anthonomus grandis grandis* Boheman (Coleoptera: *Curculionidae*) in Brazil"

_insects, 2023, doi:10.3390/insects14070582_

Round 1

Reviewer 1 Report (Previous Reviewer 2)

Authors have done a nice job adapting the manuscript. I have no further suggestions to improve the paper. 

Author Response

Thank you for your comments. Based on the current version, I am confident that the manuscript meets the necessary standards for publication in Insects.

Reviewer 2 Report (Previous Reviewer 3)

Dear Editor,

The latest version of the manuscript appears to have been significantly improved and may be accepted for publication under certain conditions:

1) To increase the impact and original contribution, I suggest the authors check whether the sex determination method proposed by Archbold Sasa et al. 2020, Sexual dimorphism in Anthonomus santacruzi (Coleoptera: Curculionidae): a biological control factor for Solanum mauritianum Scopoli (Solanaceae) Neotropical Entomology, 49, 840–850 (2020), applies to A. grandis grandis.

2) The black contour lines indicate that the cotton-growing regions in Figure 3 are almost invisible.

3) References: I don't think there is a single item prepared specifically for the Insect style!

4) Nowadays, it is a good practice to identify the studied species with molecular methods and make them available in the Genbank database. It really isn't difficult. Even with the most recognizable species, who knows…

Author Response

Reviewer#2

Thank you for your time and consideration about our manuscript. We have carefully revised our manuscript according to your valuable comments.

The latest version of the manuscript appears to have been significantly improved and may be accepted for publication under certain conditions:

1) To increase the impact and original contribution, I suggest the authors check whether the sex determination method proposed by Archbold Sasa et al. 2020, Sexual dimorphism in Anthonomus santacruzi (Coleoptera: Curculionidae): a biological control factor for Solanum mauritianum Scopoli (Solanaceae) Neotropical Entomology, 49, 840–850 (2020), applies to A. grandis grandis.

1)The method described by Sappington and Spurgeon is commonly used for determining sexual dimorphism in Anthonomus grandis. They proposed using the tergal notch as the most reliable body character for distinguishing between males and females. This method has been widely adopted in research on Anthonomus grandis grandis and is considered accurate when performed correctly.

We highlight that in the article you suggested, the technique used in the present study was mentioned in the discussion “the tergal notch was found to be the most useful body character in separating A. santacruzisexes. All A. santacruzi males (n = 65) were successfully separated from all females (n = 32) using this notch. This supports Sappington & Spurgeon (2000) that if performed correctly, the notch method of separating males and females is 100% accurate. According to the article mentioned above, distinguishing sexes in insects using the notch is easy

We modified the sentence to make it clearer that this technique was used, “Twenty-four hours after the emergence, all the weevils were separated by sex according to the description in Sappington and Spurgeon [36].” Please see lines 154-155

2) The black contour lines indicate that the cotton-growing regions in Figure 3 are almost invisible.

We modified, please see figure 3

3) References: I don't think there is a single item prepared specifically for the Insect style!

The references were corrected according to the journal's guidelines.

4) Nowadays, it is a good practice to identify the studied species with molecular methods and make them available in the Genbank database. It really isn't difficult. Even with the most recognizable species, who knows…

Thank you for the recommendations and in accordance with your comment, since the experiments have already been concluded, we were unable to use molecular methods in this study. However, we will take it into consideration and perform molecular analyses in the works that are being finalized

This manuscript is a resubmission of an earlier submission. The following is a list of the peer review reports and author responses from that submission.

Round 1

Reviewer 1 Report

overall very interesting and important study, with sound design and analysis, and easy to read. 

The authors need to reduce redundancy and unnecessary literature review (see comments) which can improve the overall quality and make the paper more concise and easier to read. It will eliminate some references, so they have to be cross-checked after revisions.

Specific comments are in the attached document

English is understandable, but some redundancy and too many words, and unnecessary information is presented. Removing these will improve overall readability and writing quality 

Reviewer 2 Report

This study provides some insights aimed at assessing the effects of constant temperatures on life characteristics of boll weevil under lab conditions. In its current condition, however, I believe this manuscript is not yet acceptable for publication in the journal of Insects. I have provided summary based on my reading of the manuscript:

1) The introduction and discussion provide no insight on how this manuscript relates to the various other ones cited in the text or concerns that have been raised by other researchers. Authors do not present any hypotheses or expectations that could be connected to previous studies (see my comments below). The authors should clearly explain why the research was done, why it was important, and how it fits with other studies.

2) My primary concern is that the authors are extrapolating the applicability of their results beyond what the design supports. These are only data from a set of seven highly artificial constant laboratory conditions (i.e., 18C-32C), so the inference power of the paper is very limited, but authors do not acknowledge this detail at all and need to be more forthcoming. Studies across a broader set of constant and fluctuating temperature regimes are, however, encouraged so that more realistic effect of temperature on biological parameters of boll weevils could be elucidated, as this is the closest to temperature fluctuations that occur in the field. This is a critical limitation of the study, and the authors must concede and discuss this. The interaction of cyclic temperatures with nonlinear characteristics of boll weevil development curves, for example, could introduce significant deviations from the results obtained in this study, and especially at the lower and higher temperatures of development functions which were not investigated at all (e.g., <18C and >32C). So, I am suggesting to the authors to tone-down the language a little and admit that there are still substantive uncertainties to be considered.

3) Some of the authors’ statements would be much stronger if they tie their work to the body of literature that has built up on the bio ecology of insect parasitoids (e.g., see https://doi.org/10.1093/jee/toz067 and https://doi.org/10.1093/jee/toy429) and insect pests (see https://doi.org/10.1093/jee/toz320). These studies provide strong evidence that daily temperature fluctuations significantly affected development times and longevity of insects studied, resulting in marked deviations and potentially erroneous predictions when compared to their constant temperature regimen counterparts. In these studies, each fluctuating temperature profile was modeled after field recorded temperatures that had the desired average target temperature. These are the first studies ever to undergo such analysis. This article should provide details on all these fronts to provide the proper context for the work.

4) My other concern is that the estimated thresholds may not equate to the biological thresholds within which boll weevil develops as only being explained by the linear model and the lack of data from the lower and higher temperatures of development functions (i.e., <18C and >32C; see my comments above). Studies on the effects of rearing temperature on insect development have been criticized because analyses commonly use single models that are considered standard to the field of investigation or are preferred for a particular taxonomic group (see https://doi.org/10.1093/aesa/saw098, or https://doi.org/10.1093/aesa/sax063 for further explanation) or are lacking enough data points (see https://doi.org/10.1093/jee/toz320 for further explanation). As a result, alternative models that could provide superior fits to experimental datasets may be overlooked. Adding these details will improve the discussion.

5) While above I have advocated for more context and connections with other research, it is also apparent that given the thin results that the paper needs to be tightened up and more efficient. Authors need to decide if the focus is on the bio ecology of boll weevils or boll weevil distribution predictions. Sure, the article would be much more compelling if the focus is on distribution predictions, but authors can only say so much based on their limited data sets (see my comments above for more details). Much of their discussion is focused on boll weevil distribution predictions; many of the details could be considered interesting but are not central to the results and should be omitted. This is not to diminish the data gathered in this study, as they are of value. But it is important for the authors not to overgeneralize, and to warn the reader, including regulatory agencies, against doing so as well.

5) Also, the discussion lacks real concluding remarks in my opinion, and if I was a practitioner or consultant, I’d want to see these recommendations for my area or city. The conclusions should also concisely summarize major findings and suggest, briefly, new avenues for research.

Overall, I was excited to see the results of the paper after reading the abstract, but I found it hard to extract key messages useful to policymakers and professionals, probably in large part due to the lack of connection with other published work and need for improved structure of the current manuscript.

Good luck!

Reviewer 3 Report

The basic part of the presented manuscript is the experiment concerning the influence of temperature on the development of boll weevils. The experiment was constructed, led, and statistically elaborated properly. The results obtained are clear and important. BUT, this experiment is in the majority a straight repetition of an almost identical experiment conducted by Shoil et al. (2005). What is more, as admitted by the authors, both experiments led to the same conclusions. In this light, there is no reason for publishing it, especially in a high-ranking journal like Insects. There are many variants of that kind of experiment (e.g., the influence of daily changes in temperature on the development of the weevil) that will really correspond to the natural condition, be welcome for publication, and be really applicable. The really new contribution presented by the authors is the attempt to compare the thermal demands of boll weevils to the temperature zones in Brazil. But, again, without many additional pieces of information like altitude, seasonal amplitude, and final daily amplitude of temperature, such a comparison seems to be only a simplification. Especially if it is presented on a very small and simple map for a country as big as Europe.

Reviewer 4 Report

This is an interesting study that reports to determine the thermal requirements of Anthonomus grandis grandis. Overall the English language requires improvement before being acceptable for publication and I have some concerns around the incubators used and how conditions were determined. Some specific comments are as follows:

Overall the simple summary needs to editing and restructuring to improve clarity.

Line 16: Change ‘as well developmental time’ to ‘as well as developmental time’

Line 17: Not sure what you mean by ‘generated a life table parameter’ better to simply say ‘We established life tables from which we calculated the net reproductive rate (R0), among other parameters.’

Line 19: Change ‘fir’ to ‘for’

Line 20: What do you mean by ‘definition of pest management strategies’?

Line 31: Change ‘better’ to ‘faster’. Also it isn’t clear that population growth was statistically significantly faster?

Line 34: Make clear that you are referring to the crop where you say ‘the most suitable areas’

Line 40: Sentence starting on this line needs to be rephrased to make clear why this is an economically important pest. Also, avoid simply saying ‘major’ as this is too vague.

Line 41: Clarify what you mean by ‘Since its entry to Texas in the 1890s’

Line 45: The sentence starting on this line needs to be rephrased as it reads as though cotton is grown in these regions because the environment favours insects?

Line 56: Please rephrase the sentence ‘As a result, local cotton crops have suffered significant damage becoming the cotton pest that drives the integrated pest management decision with the most insecticide applications requiring frequent applications of insecticides to control the pest. [8, 9, 10].’ The sentence doesn’t currently make sense and you are trying to say a lot in the one sentence.

Line 60: The sentence starting on this line indicates losses per hectare and throughout Brazil but there is no indication of the area of cotton grown in the country. In addition, ‘control costs and depreciation of pest-control infrastructure’ are costs and not losses. This needs to be made clear.

Line 62: What do you mean by ‘the major cotton pest’?

Line 67: Can males even enter reproductive diapause? Please provide evidence.

Line 69: The sentence on this line needs to be reworked to better describe day degrees and insect development.

Line 82: Why is it essential to rear insects on artificial diets for pest control programmes? This doesn’t seem to make any sense as it is written. Why not grow crops under artificial conditions instead? This whole paragraph is opinion and not supported by any references.

Line 92: Change ‘conducted’ to ‘completed’

Line 96: How does cotton meal and cotton meat differ?

Line 97: Name and give some details of what the most used diet is.

Line 112: Is this diet a standard diet previously published?

Line 119: How can you have two screens on the base of the cage, this needs to be better explained. Also give details of the mesh used and supplier details of the cages.

Line 125: What size Petri dish and were these ventilated? How much diet provided? What were the environmental conditions?

Line 135: Give manufacturer details of climate chambers. How were conditions inside each chamber independently verified? Did you record humidity inside Petri dishes or well plates as this would have differed to the 70% in the chamber.

Line 139: Give details of the Petri dishes again and how much diet provided.

Line 143: Give supplier details of the plates used.

Line 144: How did you select which 24 insects from each Petri dish of 50 eggs to use for larval development? How much diet was placed in each well? How were larvae prevented from escaping?

Line 146: Surely sex ratio was determined by the larvae selected to rear through and not by the sex ratio produced by ovipositing females?

Line 150: Do you know that this adaptation prevented humidity from increasing or is this assumed?

Line 152: When you say ‘they were replaced daily’ do you mean the insects or the diet?

Line 188: Checked formatting for ‘R0’

Line 200: Give the pupal duration and not just the difference.

Line 207: Change ‘Larval’ to ‘Larvae’

Line 222: What do you mean by ‘lower energy expenditure’?

Line 230: Table legend is on the wrong page. Also, no mention as to why some text in the table is in bold?

Line 236: Rephrase ‘suitable temperature’

Line 247: Avoid single sentence paragraphs.

Line 250: Change ‘best’ to ‘highest’

Line 260: Why is this text below the table and not within the legend? What does ‘R_0’ mean?

Line 264: This section would be improved if there was some mention of pest status of this insect in different regions of Brazil. Your model could be tested by what growers find on the ground within crops.

Line 278: What do you mean by ‘greater population growth’ here?

Line 280: What do you mean by ‘development of the pest’ here? Are you referring to development of the individual insect or of the pest status of the species?

Line 284: The scale in this figure needs to be more clearly explained.

Line 295: Change ‘of the insect’ to ‘of this species’

Line 296: Do you mean ‘different temperature conditions’ or ‘different geographic regions’ or both?

Line 298: It isn’t ‘vulnerability’ but rather simply reduced survival at these temperature extremes.

Line 304: Change ‘The better fecundity performance were found’ to ‘Higher fecundity was found’

Line 307: I don’t understand this line e.g. ‘where also detrimental to oviposition’?

Line 309: This is a very long and slightly jumbled sentence, try to shorten or break into shorter sentences.

Line 314: What do you mean by ‘boll 314 weevil for tropical origin (Brazil).’?

Line 317: Provide some supporting evidence for this statement.

Line 318: This is a tool but it is for others to decide if it is ‘interesting’

Line 331: Make sure species name is italicised.

The English language needs to be addressed. There are many ambiguous statements throughout the manuscript that must be addressed.